# Ochratoxin A-Induced Nephrotoxicity: Up-to-Date Evidence

**DOI:** 10.3390/ijms222011237

**Published:** 2021-10-18

**Authors:** Chong-Sun Khoi, Jia-Huang Chen, Tzu-Yu Lin, Chih-Kang Chiang, Kuan-Yu Hung

**Affiliations:** 1Graduate Institute of Toxicology, College of Medicine, National Taiwan University, Taipei 106, Taiwan; d05447005@ntu.edu.tw (C.-S.K.); f04447010@ntu.edu.tw (J.-H.C.); 2Department of Anesthesiology, Far-Eastern Memorial Hospital, New Taipei City 22060, Taiwan; 3Department of Internal Medicine, College of Medicine and Hospital, National Taiwan University, Taipei 106, Taiwan; kyhung@ntu.edu.tw; 4Department of Integrated Diagnostics & Therapeutics, National Taiwan University Hospital, Taipei 10002, Taiwan

**Keywords:** ochratoxin A, nephrotoxicity, molecular interaction, prevention

## Abstract

Ochratoxin A (OTA) is a mycotoxin widely found in various foods and feeds that have a deleterious effect on humans and animals. It has been shown that OTA causes multiorgan toxicity, and the kidney is the main target of OTA among them. This present article aims to review recent and latest intracellular molecular interactions and signaling pathways of OTA-induced nephrotoxicity. Pyroptosis, lipotoxicity, organic anionic membrane transporter, autophagy, the ubiquitin-proteasome system, and histone acetyltransferase have been involved in the renal toxicity caused by OTA. Meanwhile, the literature reviewed the alternative or method against OTA toxicity by reducing ROS production, oxidative stress, activating the Nrf2 pathway, through using nanoparticles, a natural flavonoid, and metal supplement. The present review discloses the molecular mechanism of OTA-induced nephrotoxicity, providing opinions and strategies against OTA toxicity.

## 1. Introduction

Ochratoxin A (OTA) is a mycotoxin frequently found in various foods and animal feed [1,2,3,4]. OTA has nephrotoxic deleterious effect on human health due to exposure to OTA-contaminated food [5]. Previous studies have confirmed that exposure to OTA can lead to multiorgan toxicity [6,7,8], but the kidney is the primary target of OTA [9]. There is a hypothesis about Balkan nephropathy correlated with OTA [10]. Still, subsequent research showed no statistically significant evidence for the connection between OTA exposure and human health risks in the Bulgarian or Croatian study populations [11]. However, a preliminary study in Egypt found that OTA may correlate to renal disease by high serum OTA level in end-stage renal disease (ESRD) patients, high serum and urine OTA level in nephrotic syndrome, and urothelial cancer [12]. Hsieh et al. found that daily excretion of OTA is correlated with 24 h urine protein excretion, which implied OTA might lead to proteinuria [13]. The animal study showed that chronic kidney disease (CKD) dogs had higher plasma OTA concentrations than healthy dogs. Still, there is no significant correlation between creatine and OTA plasma level in CKD dogs [14]. Kosicki et al. demonstrated the same result, that plasma OTA had been detected in dialysis patients and healthy groups, but no significant difference between these two groups [15]. The most direct evidence of OTA’s effect on the kidney is that OTA was detected in human kidney tissue. Ostry et al. demonstrated that OTA was detected in human renal tissue with a level range from 0.1–0.2 ug/kg and a mean of 0.07 ug/kg [16]. In addition, a human kidney from nephrectomy showed a concentration of OTA that was 0.15–0.39 ng/kg [17]. After absorption, OTA could be excreted by the urine, feces, and milk [18]. Urine is a non-invasive biomonitor, but ingested OTA is excreted in the urine in only a small amount [19]. OTA was detected in urine samples from Germany, Haiti, and Bangladesh. The median level of OTA was 0.097, 0.038, and 0.115 ng/mL, respectively [20]. Ali et al. collected urine samples in German adults to investigate OTA and its metabolite ochratoxin alpha (OTα) in plasma and urine. The urinary level of OTA and OTα was 0.21 ± 0.31 ng/mL (mean), 1.33 ± 2.63 ng/mL (mean), respectively; it suggested OTα could be another biomarker for monitoring uptake of OTA [21]. Ochratoxin A-8-β-glucuronide was the product of glucuronides of OTA in urine. Munoz et al. collected urine from infants and adults to investigate glucuronides of OTA as total OTA in urine. This study showed in German infants, the mean OTA-aglycone concentration was 80 ± 14 ng/L, and OTA-total was 98 ± 114 ng/L after hydrolysis in German infants; in Turkish infants, the mean OTA-aglycone concentration was 285 ± 57 ng/L, and the mean value for OTA-total was 1041 ± 275 ng/L, same as a result in a German adult, urinary OTA levels were 76 ± 29; the OTA-total was 205 ± 114 ng/L, which implied that Ochratoxin A-8-β-glucuronide could be another biomarker to avoid underestimate OTA exposure [19]. Silva et al. collected 85 urine samples from Portuguese children; 92.94% were positive for OTA detection. The mean OTA concentration level was 0.020 ng/mL with a maximum 0.052 ng/mL value. The risk assessment by probable daily intake (PDI) was 10 to 194% [22]. Thus far, OTA has been detectable in infant cereal, breakfast cereal, and dried fruits. Infant and children are the highest OTA exposure population in the US, but there is no significant risk of adverse effects after OTA exposure [23]. Huong et al. found that the dietary exposure to ochratoxin of Vietnamese children was 52.6 ng/kg/day. Rice, egg, milk, and beans are the most commonly OTA-contaminated food. The population has a risk of renal cancer, which was evaluated by an age-adjusted margin of exposure (MOE) in this study [24]. In addition, the occurrence of OTA in a study in Poland demonstrated that low and constant OTA levels present in animal tissue and feed, implying continued exposure of OTA to animal and human beings [25].

## 2. The Mechanisms Underlying OTA Nephrotoxicity

The mechanisms of OTA-induced nephrotoxicity include inhibition of protein synthesis, DNA damage, cell cycle arrest, and cell apoptosis [26]. OTA is recognized as a strong nephrotoxin on its accumulation in proximal tubule epithelial cells and initiating cellular damage through oxidative stress, DNA damage, apoptosis, and inflammatory response [27].

### 2.1. Oxidative Stress

OTA induces oxidative stress in rat kidneys and liver by increasing MDA, lipid peroxidation, and suppressing glutathione, catalase, and superoxide dismutase [28]. In addition, OTA induced reactive oxygen species (ROS) formation in renal proximal tubular cells and increased formation of 8-oxoguanine-related DNA damage; N-acetylcysteine (NAC) treatment decreased ROS level and increased cell viability after OTA exposure [29]. OTA enhances apoptosis signal-regulated kinase 1 (ASK-1) expression, which regulates ROS production and decreases mitochondrial membrane potential to promote nephrotoxicity after OTA exposure [30]. OTA activates aryl hydrocarbon receptor (AhR) and pregnane X receptor (PXR) signaling pathways to induce cytochrome P450 1A1 (CYP1A1), CYP1A2, and CYP3A4, leading to increased ROS production. ROS enhanced Nrf2 translocated to the nucleus to increase expression of heme oxygenase-1 (HO-1), glutamate-cysteine ligase catalytic subunit (GCLC), and NADPH quinone oxidoreductase 1 (NQO1) to ameliorate cell damage. Therefore, modulation of AhR, PXR, and Nrf2 could be considered to prevent or treat OTA-induced toxicity [31].

### 2.2. Nitrosative Stress

OTA-induced inducible nitric oxide synthase (iNOS) expression and protein nitration eventually lead to nitrosative stress in kidney and liver cells [32]. Crupi et al. showed OTA increases nitric oxide (NO) production in different kinds of kidney cell lines [33]. Longobardi et al. revealed that OTA promoted iNOS expression and increased NO production during in vitro study [34]. In contrast, Damiano et al. found OTA reduced NO production in rat kidneys, which may be related to OTA-induced hypertension by inhibiting NO synthase (NOS). Besides, OTA enhanced superoxide production which inhibited sodium-hydrogen exchanger isoform 3 (NHE3) and lead to disturbance of fluid reabsorbtion. Therefore, alteration of proximal tubular reabsorption by OTA-induced ROS would affect balance between fluid and solute [35].

### 2.3. Apoptosis/Pyroptosis

OTA induced apoptosis of renal cells with activation of caspase-3 and increased DNA fragmentation [36,37]. Besides, a low dose of OTA induced oxidative stress and stimulated apoptosis of rat renal tubular cells [38]. OTA induced necrosis and apoptosis of human proximal tubular epithelial cells (HK-2) with increasing caspase 3/7 activities and cleaved PARP. In addition, OTA induced apoptosis in HK-2 cells by activating the MEK/ERK 1/2 signaling; meanwhile, OTA activated c-MET/PI3K/AKT pathway for antiapoptotic survival signaling [27]. Li et al. revealed p53 induced by OTA could protect kidney epithelial cells through suppression of the JNK-related apoptotic pathway [39]. Sheu et al. found OTA promotes ER stress and apoptosis through activation of calpain; meanwhile, OTA induced NADPH oxidase expression, which regulates ROS production in renal mesangial cells [8].

OTA attenuated cholesterol and sphingomyelin to disrupt lipid raft formation, which regulates the apoptosis of HK-2 cells through regulating the PTEN/PI3K/AKT pathway [40]. Transcriptomic analysis found that hypoxia, epithelial-to-mesenchymal transition (EMT), apoptosis, and the xenobiotic metabolism pathway are affected by OTA. The following result showed OTA-induced EMT, apoptosis, and kidney injury by regulating the AhR-Smad3/HIF-1α signaling pathway. This study provided an opinion against OTA-induced renal toxicity through understanding of OTA-induced EMT and apoptosis-related molecular pathway [41]. Zhang et al. discovered tumor necrosis factor receptor-associated protein 1 (TRAP-1) enhanced expression of Bcl-2 and inhibited expression of CypD, Bax, GRP78, and CHOP against apoptosis. OTA could inhibit TRAP-1, leading to mitochondria-mediated and ER stress-excited apoptosis of proximal tubular cells [42]. OTA could induce apoptosis of HK-2 cells through enhancement of ERK 1/2 phosphorylation by activating the NF-kB pathway [43]. Another study found OTA induced nephrotoxicity and apoptosis through the p38 pathway in porcine kidney epithelial cells (PK 15); meanwhile, OTA induced immunotoxicity through the ERK signaling pathway in porcine primary splenocyte. This study clarified that OTA induced toxicity through different molecular pathways in different cells [44]. Qi et al. discovered organic cation transporters 2 (OCT2) participate in OTA-induced cell apoptosis in rat proximal tubule cells (NRK-52E) cells by increase caspase 3 and cyclin dependent kinase 1 (CDK1), but a cell transport experiment would be conducted to ascertain the role of OCT2 in OTA-induced toxicity [45]. Besides apoptosis, OTA induced nephrotoxicity via activating NOD-like receptor protein 3 (NLRP3) inflammasome, and caspase1-dependent pyroptosis lead to renal fibrosis by evidence of increasing TGF-β and α-SMA during in vivo and in vitro study. This study offers a novel opinion or strategy for the treatment of OTA-induced renal fibrosis [46].

### 2.4. DNA Damage/Genotoxicity

OTA promoted oxidative DNA damage in the liver and kidney of rats [47], but OTA caused DNA damage without lipophilic DNA adduct formation [48]. OTA-induced delete mutation of homologous repair of DNA double-strand break lead to genotoxicity; p53 could suppress deletion mutation of homologous repair after OTA exposure. In addition, p53 attenuated OTA-induced karyomegaly and apoptosis in mice kidneys [49]. OTA could produce genotoxicity through different bioactivation. OTA quinone (QTQ) is the product of OTA after oxidation of CYP450; QTQ could be reduced by ascorbate to form OTA hydroquinone (QTHQ). Besides, dehalogenation of OTA by a reductive process lead to production of a reactive aryl radical which could react at the purine C8 site to form DNA adduct [50]. OTA produced C-C8 and O-C8 OTA-3′-dGMP covalent DNA adducts through photooxidation with 3′-dGMP; these DNA adducts were found in rat kidney and pig kidney after chronic and subacute exposure of OTA [51]. Mantle et al. also demonstrated that C-C8 OTA 3′dGMP in rat kidneys directly binds OTA to DNA after OTA exposure [52]. OTA activated the DNA methyltransferase 1 (DNMT1) dependent JAK2/STAT3 signaling pathway to induce apoptosis and DNA damage in PK15 cells and porcine alveolar macrophage (PAM); meanwhile, OTA increased the expression of suppressors of cytokine signaling 3 (SOCS3), which was regulated by DNMT1 only in PK15 cells. This study elucidated that OTA induced nephrotoxicity instead of immunotoxicity through the DNMT1/JAK2/STAT3/SOCS3 signaling pathway [53].

### 2.5. Epigenetic Modification

Ozden et al. found that OTA-induced hypermethylation and hypomethylation on the different genes of rat kidneys involved in the mammalian target of rapamycin (mTOR) signaling pathway may contribute to renal carcinogenesis [54]. OTA induced global methylation in rat kidneys through enhanced expression of DNMT1 and DNMT3b; hypermethylation of E-cadherin and N-cadherin by OTA exposure results in decreased expression E-cadherin and N-cadherin, which may involve in Wnt and PI3K/Akt signaling pathways [55]. OTA activated DNMT1 and histone deacetylase1 (HDAC1) to induce cytotoxicity and apoptosis through ROS production [56]. Another study discovered inhibition of histone acetyltransferase (HAT) by OTA, leading to loss of histone 3 lysine 9 (H3K9) acetylation to suppress gene expression. This study disclosed loss of histone acetyltransferase is a critical step to OTA-induced carcinogenesis, but alteration of non-histone acetylation by proteome-wide analysis should be conducted to clarify the relationship between HAT inhibition and OTA-mediated gene expression change, mitosis, and DNA repair [57]. In addition, OTA induced cell cycle arrest in the G0/G1 phase of HK-2. Some of the gene was related to G0/G1 cell cycle arrest, but only increased expression of PTEN due to OTA-induced hypomethylation of PTEN which involved in G1/S phase transition. G0/G1 phase cell cycle arrest and inhibition of the Notch and Ras/MAPK/CREB signaling pathway after OTA exposure would lead to renal toxicity [58]. 

### 2.6. Inhibition of Protein Synthesis

From the previous study, OTA inhibited protein synthesis in various organs, including the spleen, liver, and kidney, which could be prevented by phenylalanine; this implied that OTA competes with phenylalanine leading to inhibition of phenylalanine-tRNA synthase [59]. Besides, OTA could inhibit phenylalanine hydroxylase, which could be reduced by phenylalanine [60]. Argawal et al. used molecular docking analysis to reveal that OTA could bind on a common pocket of phenylalanine-tRNA synthase to disrupt protein synthesis [61]. 

### 2.7. Cell Cycle Arrest

OTA disturbed the cell cycle by increasing cell count at G0/G1phase and G1/G2 phase, decreasing cell count at the S phase [62]. OTA inhibited cell cycle progression in the S phase with decreased expression of cyclin A2, cyclin E2, and cyclin-dependent kinase 2 (CDK2) while inducing DNA damage in HEK 293 cells [63]. N-acetylcysteine (NAC) could reverse OTA-induced cell cycle arrest by enhancing cyclin A2, cyclin E2, and CDK2 expression, and also alleviates OTA-induced ROS production and DNA damage [64]. Recent studies discovered OTA-induced cell cycle arrest in the G1 and G1/S phase mediated by p53 through suppression of cyclin D1, Cdk2, and Cdk4 in HK-2. This study suggested the status of gene and protein should be showed after being regulated by an upper mediator such as p53, because many genes, proteins, intermediate molecular, and even upper mediators could regulate cell cycle [65]. Using weight correlation network analysis by Duborg et al. found that CDK2 plays a major regulator in G1 cycle arrest induced by OTA. OTA inhibit expression of CDK2, overexpression of CDK2 attenuated OTA-induced G1 phase cycle arrest. [66].

### 2.8. Lipotoxicity

As evidenced by lipid deposition, OTA-induced renal lipotoxicity was found in the renal cortex and medulla after OTA exposure. In addition, OTA induced triglyceride levels elevation in the kidney. Moreover, OTA suppressed the expression of peroxisome proliferator-activated receptor alpha (PPAR-α) and medium-chain acyl-CoA dehydrogenase (Mcad), which is needed for fatty acid oxidation. Meanwhile, OTA promoted lipid peroxidation and sphingomyelinase, which might be involved in apoptosis. Besides, OTA inhibited activity of brown adipose tissue (BAT) to disturb the energy metabolism balance [67]. 

### 2.9. Mitochondrial Dysfunction/ATP

Mitochondrial membrane potential plays an essential role in mitochondrial homeostasis. The transmembrane potential of hydrogen ion that was formed by mitochondrial membrane potential is used to make ATP. A rise or decrease in mitochondrial membrane potential will affect cell viability [68]. Argawal et al. found loss of mitochondrial membrane potential at the initiation of OTA treatment (4–12 h), followed by hyperpolarization of mitochondrial membrane potential at 24 h [61]. Eder et al. discovered that OTA induced hyperpolarization of mitochondrial membrane potential and increased cellular ATP [69]. In contrast, Chebotareva et al. found OTA decreased intracellular ATP, which can be reversed by Diosmetin (DIOS) [70]. In addition, OTA inhibited mitochondrial oxygen consumption through suppressed complex I and II of the mitochondrial respiratory chain, implying that OTA caused mitochondrial dysfunction in rat renal proximal tubule [71]. Heat shock proteins (HSP) play an intracellular defensive role in the kidney, which promotes the ability against apoptosis and necrosis [72]. Glucose-regulated protein 75 (GRP-75) belongs to the HSP70 family, it may protect the renal cell from mitochondrial dysfunction through regulating Lon protease 1 (Lonp1) after OTA exposure. Besides, decreased expression of GRP75 was concurrent with an increasing OTA dose, thus GRP-75 could be a biomarker of renal tubular necrosis induced by foodborne toxicity such as OTA [73].

### 2.10. miRNA

miRNA played a regulatory role in the development of nephropathy, renal carcinoma, and urothelial cancer. Upregulation of miR-497, miR-133a-3p, miR-423-3p, miR-34a, miR-542-3p, and downregulation of miR-421-3p, miR-490, and miR-9840-3p after OTA exposure in piglets thus altered miRNA link to a different cellular pathway and could be a biomarker of OTA exposure [74]. OTA decreased miR-29b expression led to promoting the expression of collagen protein, which has the potential for developing renal fibrosis [75]. Stachurska et al. found that OTA increased miR-132, miR-200 by ROS, and TGF-β to inhibit Nrf-2, HO-1 which led to suppressed proliferation and cell viability in renal proximal tubular epithelial cells [76].

### 2.11. Autophagy

A recent study revealed OTA-activated autophagy promoted cell apoptosis. In addition, OTA induced ubiquitin-proteasome system (UPS) and promoted degradation of ubiquitinated protein through 26S pure proteasome. Meanwhile, OTA increased PI3K/AKT and MAPK/ERK1-2 signaling pathways by degradation of phosphatases dual specificity phosphatase 3 (DUSP3), dual specificity phosphatase 4 (DUSP4), and Ph domain and Leucine-rich repeat protein phosphatase (PHLPP) through autophagy and UPS, which may be involved in OTA toxicity and carcinogenicity [77]. However, Qian et al. found autophagy could protect PK cells from OTA-induced apoptosis by suppressing the Akt/mTOR signaling pathway. This study provided a novel therapeutic agent against OTA toxicity through autophagy [78].

### 2.12. EMT/Fibrosis/Tight Junction

An epithelial-to-mesenchymal transition (EMT) is involved in renal fibrosis, and an experiment discovered OTA activated TGF-β/Smad2/3 and B-catenin/Wnt signaling pathways to induce EMT and renal fibrosis by evidence of increasing expression of α-SMA and fibronectin. It is possible to prevent OTA-induced EMT related renal fibrosis through regulating TGF-β/Smad2/3 and B-catenin/Wnt pathways [79]. Another study found activation of the ERK1/2 NF-kB signaling pathway by OTA exposure induced glomerular inflammation and fibrosis [80]. OTA disturbed intercellular interaction by disrupting the tight junction by suppressing occlusion and ZO1 in MDCK cells [81].

### 2.13. Other Mechanisms

OTA induced inflammation of human embryonic kidney cells (HEK293) through the NF-kB pathway; meanwhile, as OTA concentration increased, it shifted the inflammatory environment to a pro-apoptotic environment [82]. OTA induced ER stress and autophagy; meanwhile, it promoted phosphorylation of ERK 1/2 and p38 of kidney and spleen in pigs [83]. Besides, OTA promoted senescence in human renal proximal tubular cells by increasing senescence-associated-β-galactosidase (SA-β-gal), senescence-associated secretory phenotype (SASP), DNA damage, cell spreading through activating p53-p21, p16-pRB signaling pathway and a disturbed cell cycle. OTA-induced senescence may provide a condition to become cancer cell, and therefore understanding of the senescence mechanism would help to study OTA-induced carcinogenesis [84]. Besides, increasing evidence found that heme oxygenase-1 (HO-1) has a cytoprotective role in renal disease [85]. OTA induced HO-1 to protect porcine kidney cells from mitigating inflammation, fibrosis, and apoptosis produced by OTA. Meanwhile, HO-1 regulated the expression of Nrf-2, miR-21, miR-29b, and p53-regulated miR-34a after OTA exposure. In addition, this study suggests further experiments could be conducted to clarify the role of HO-1 in OTA-induced genotoxicity [86]. Exposure to OTA may increase angiogenesis by increasing the expression of HIF-1α and EPO gene after 24 h; meanwhile, expression of TGF-β and vascular endothelilal growth factor (VEGF) may increase significantly at 48 hours. OTA may interrupt the normal metabolic process of thorough alteration of ATP production and pyruvate dehydrogenase 1(PDK1) expression. HIF-1α regulates cellular adaptation to hypoxia and hypoxic condition may promote carcinogenesis, therefore investigating OTA-induced hypoxic condition may help to clarify OTA related carcinogenesis. Besides, hypoxic-affected ATP production and the transformative capability of OTA could be investigated in further experiments [87]. Loboda et al. found that OTA-induced nephrotoxicity is sex-dependent, with males more susceptible, and regulated by the Nrf-2 pathway. After OTA exposure, it increased renal fibrosis and decreased the expression of glutathione reductase (GR), superoxide dismutase (SOD), claudin-2, and VEGF in male Nrf-2 knockout mice. Nrf2 deficiency promoted susceptibility to OTA-induced renal injury, thus modulation of Nrf2 provides a therapeutic option against OTA-induced renal disease [88]. Imaoka et al. used a 3D human kidney proximal tubule microphysiological system (kidney MPS) to validate OTA dose–response toxicity. Cell death is induced by OTA toxicity in a concentration-dependent manner. Besides, OTA is metabolized by P450(s) and OTA caused transcriptional suppression of glutathione S-transferase (GST) enzymes, which lead to nephrotoxicity by attenuate detoxification or conjugation of glutathione. Meanwhile, organic anionic membrane transporter(s) are involved in the accumulation of OTA in the proximal tubule by evidence of intracellular OTA accumulation after organic anionic membrane transporter inhibitor (probenecid) was used. However, application of MPS has a technical limitation to assess toxicology, such as limited cell number and lack of throughput, and is limited in cell sorting [89].

Table 1 and Table 2 summarize recent studies of OTA-induced nephrotoxicity.

## 3. Prevention

Continued exposure of OTA to human beings and animals has been detected in a previous study [25], therefore different kinds of strategies are used to protect humans and animals from OTA toxicity through understanding the molecular mechanism. 

### 3.1. Antioxidant

#### 3.1.1. Ursolic Acid (UA)

OTA promoted ROS production and enhanced renal cell death by suppressing Lonp1, aconitase 2 (Aco2), and heat shock protein 75 (Hsp75). In addition, ursolic acid could reverse the expression of Lonp1, Aco2, and Hsp75 to ameliorate OTA-induced toxicity and ROS production in renal cells. During different kinds of UA treatment, pretreatment of UA is most effective to ameliorate OTA toxicity [90].

#### 3.1.2. Hydroxytyrosol

Hydroxytyrosol is a phenolic compound with antioxidant, antimicrobial, antitumoral, cardioprotective, and neuroprotective properties. Pretreatment of hydroxytyrosol prevented OTA-induced renal injury by reducing ROS level, nitrite production, kidney fibrosis in vivo and in vitro. This study provided evidence that a phenolic compound could against oxidative stress related renal injury [33]. 

#### 3.1.3. N-Acetyl-L-tryptophan (NAT)

NAT reduced neuroinflammation, oxidative stress, and apoptosis [91]. Pretreatment of N-Acetyl-L-Tryptophan (NAT) could bind on phenylalanine tRNA synthetase may decrease OTA-induced protein inhibition; in addition, NAT also protected HEK cells from ROS insult, cell cycle arrest, disturbed mitochondrial membrane potential induced by OTA. Prophylactic dietary application of NAT may prevent OTA-induced toxicity [61]. 

#### 3.1.4. Troxerutin

Troxerutin is a natural flavonoid from rutin that has anti-inflammation and antioxidant properties. Cotreatment of troxerutin mitigate lipid accumulation in renal cells, relieved lipid peroxidation, and reversed the level of PPAR-α, medium-chain acyl-CoA dehydrogenase (Mcad) and sphingomyelinase (SMase) to alleviate OTA-induced renal lipotoxicity. In addition, Troxerutin improved whole-body energy metabolism by attenuating OTA-suppressed brown adipose tissue (BAT) activity [67]. 

#### 3.1.5. Taurine

Taurine is an amino acid involved in membrane stabilization, antioxidation, immunomodulation, osmoregulation. Taurine alleviated apoptosis by decreasing expression of Bax and caspase 3, and inhibited autophagy through decreased expression of light chain 3 (LC3) induced by OTA in PK15 cells. However, the role of taurine in between apoptosis and autophagy induced by OTA remains unclear; taurine may regulate autophagy to maintain cellular homeostasis after OTA exposure. This study suggest taurine has potential against other mycotoxin or environment-induced toxicity [92].

#### 3.1.6. δ-Tocotrienol (Delta)

δ-tocotrienol is a natural vitamin E that reverses antioxidant enzymes superoxide dismutase (SOD), catalase (CAT), and glutathione (GSH) activity to reduce OTA-induced oxidative stress. Meanwhile, cotreatment of Delta suppressed OTA-induced superoxide production, elevated blood pressure, and decreased glomerular filtration rate (GFR). This study demonstrated that Delta is a ROS scavenger and the dietary strategy has potential to counteract OTA toxicity [93]. 

#### 3.1.7. Curcumin

Curcumin is natural polyphenolic compound produced from the Curcuma longa, and it alleviates oxidative stress, inflammatory response, and activates the Nrf2 pathway against kidney injury [94]. Cotreatment of curcumin restores glutathione peroxidase and against OTA-induced glomerular damage, tubular damage, tubulointerstitial fibrosis, and inflammation in vivo [95]. Longobardi et al. demonstrated that cotreatment of curcumin attenuated OTA-induced nitrosative stress and inflammatory and DNA damage by reducing NO production, iNOS expression, TNF-α, IL-1B, IL-6, and 8-Hydroxy-2′-Deoxy Guanosine (8-OHdG) in the kidneys and liver of rats [34].

#### 3.1.8. Yemeni Green Coffee/Red Orange and Lemon Extract (RLE)

Cotreatment of Yemeni green coffee suppressed OTA-induced oxidative stress by restoring SOD and glutathione; it reduced rat kidney injury under histopathology. Yemeni green coffee is a functional food which has antioxidant properties against OTA-induced toxicity [96]. RLE is rich in cyanidin 3-glucoside antioxidant. Cotreatment of RLE reversed GSH level and reduced OTA-induced glomerular damage, tubular damage, inflammation, renal fibrosis, and oxidative stress. RLE is vegetable-based diet which could be added to feed to protect humans and animals from OTA-contaminated food [97]. 

#### 3.1.9. Recombinant Mitochondrial Manganese Containing Superoxide Dismutase (rMnSOD)

Ciarcia et al. demonstrated that cotreatment of recombinant mitochondrial manganese-containing superoxide dismutase (rMnSOD) restored blood pressure, GFR, and attenuated morphologic change and interstitial fibrosis in rat kidneys after OTA treatment [98]. In addition, cotreatment of rMnSOD increases the production of nitric oxide (NO), restores impaired sodium–hydrogen exchanger isoform 3 (NHE3) regulating fluid reabsorption in the proximal tubule, and decreases blood pressure after OTA exposure in rat kidneys [35]. 

#### 3.1.10. Astaxanthin (ASX)

Astaxanthin (ASX) is an antioxidant that improves bisphenol A (BPA)-induced impaired mitochondrial function in rat kidneys [99]. Li et al. demonstrated that pretreatment of ASX activated the Nrf2/Keap1 signaling pathway to protect rat kidneys from OTA-induced oxidative damage. The antioxidant ability of ASX could provide protection from other mycotoxin-induced damage [100]. 

### 3.2. Nrf2 Activator

#### 3.2.1. Luteolin (LUT)

LUT is a dietary flavonoid that has antioxidant properties from recent reports, such as that LUT restored autophagy though the AMPK pathway to mitigate mercury-induced destruction of renal tubule, inflammation, oxidative stress, and apoptosis of rats’ kidneys [101]. LUT alleviated lead-induced kidney injury by inhibiting oxidative stress, inflammation, apoptosis, and enhanced expression of Nrf2, HO-1 in an in vitro study [102]. Liu et al. have found that cotreatment of LUT promoted expression of Nrf2 and antioxidant enzyme to ameliorate ROS level and mitochondrial membrane potential due to OTA exposure in rat kidney cells. In addition, LUT activated HIF-α to regulated angiogenesis and epithelial restitution process. These results suggest LUT is a potential dietary flavonoid to counteract other mycotoxins or environmental toxins [103].

#### 3.2.2. Sulforaphane (SFN)

Sulforaphane (SFN) is an Nrf2 activator; it restored OTA-suppressed Nrf2 transactivate of antioxidant response element (ARE) in porcine kidney tubular cells [104]. Loboda et al. found that pretreatment of SFN enhanced Nrf2, HO-1, NQO1 expression; it also reduced inflammatory cytokine IL-1B, IL-6, pro-apoptotic factor (c-myc, puma), miR-382, and miR-34a after OTA exposure [88]. 

### 3.3. Trace Element

#### 3.3.1. Selenium 

Selenium is an essential trace element with antioxidant, anticancer, and immune modulation against mycotoxin-induced oxidative toxicities [101]. Cotreatment of selenium yeast (Se-Y) promoted PI3/AKT and Nrf2/Keap1 signaling pathways to alleviate OTA-induced oxidative damage and apoptosis in the kidney of chickens. This study indicate that Se-Y has an antiapoptotic and antioxidant effect to prevent OTA-induced renal injury in chickens [102]. OTA downregulated the expression of glutathione peroxidase (GPx1), leading to DNMT1 mediated DNA damage during in vitro study; selenium reversed OTA-induced DNA damage by increasing DNA repair gene expression and inhibited accumulation of gamma-H2AX foci through restoring expression of GPx1and SOCS3. Pretreatment of selenium could enhance GPx1 to protect PK15 cells from OTA-induced DNA damage [103]. Selenoprotein S (SelS) has antioxidant properties; it alleviated the OTA-induced apoptosis by suppressing phosphorylation p38, meanwhile, SelS reduced ROS production after OTA exposure in PK15 cells [105]. Pretreatment of selenium probiotics (SP) also attenuated OTA-induced piglet kidney injury through enhanced expression of GPx1, SOD, and inhibited DNMT1 [106]. Accumulation of this evidence found that selenium could diminish OTA-induced renal injury by antioxidant, antiapoptosis and reduced DNA damage, therefore selenium could be a feed additive to counteract mycotoxin contaminated food.

#### 3.3.2. Zinc

Trace elements are components of intracellular antioxidants and can maintain cellular homeostasis. Lycopene restored OTA inhibition of selenium, zinc, and copper in the kidney, liver, and testis of rats [107]. In addition to being a component of many important enzymes, Zinc can also regulate the antioxidant gene expression. Li et al. found that pretreatment of zinc supplements promoted metallothionein (MT) expression to reduce ROS production and apoptosis in kidney epithelial cells after OTA exposure [108]. Besides, zinc supplements reduced ROS production and increased expression of SOD; furthermore, zinc supplement maintained DNA stability by reducing DNA strand break, 8-OHdG formation and DNA hypomethylation after OTA exposure in HepG2 cells [109]. Additionally, Bodiga disclosed that zinc replenishment suppressed intracellular superoxide production, apoptosis, and attenuated ER stress in hypoxia/reoxygenation of cardiomyocytes [110]. Similar to selenium, Zinc acts as a ROS scavenger, maintains DNA stability and it has an antiapoptotic property against OTA-induced toxicity. 

### 3.4. Nanoparticle

Cotreatment fed with nanoparticles of hydrated sodium aluminum silicates or copper oxide reduces fish mortality and alleviates OTA-induced inflammation, tubular degeneration, and necrosis in the kidney of fish. These nanoadsorbents are not absorbed in the gastrointestinal tract due to it binding with OTA to become a nanoadsorbent-OTA complex which is excreted from the body [111]. Abdel et al. found cotreatment of chitosan nanoparticle (CSNP) or grafting of gallic acid on CSNP attenuated oxidative stress, DNA fragmentation, and histopathology change caused by OTA in catfish [112]. The others demonstrated that cotreatment of chitosan nanoparticles alone or combined with quercetin alleviated OTA-induced oxidative stress and apoptosis in vivo through the ROS scavenger property of chitosan nanoparticle and enhancement of quercetin antioxidant capacity [113]. These recent studies of nanoparticles provide another candidate to attenuated OTA-induced toxicity. 

Recent studies of prevention of OTA-induced nephrotoxicity are summarized in Table 3.

## 4. Conclusions

The mechanisms of OTA-induced nephrotoxicity are complicated and incompletely clarified; the literature review found that OTA activates PTEN/AKT, ERK1/2, NF-kB, and p38 pathways to induce apoptosis. Meanwhile, OTA induced lipotoxicity, pyroptosis, and senescence of renal cells. Besides, OTA regulates transcriptional factors, DNA methylation, and histone medication to cause renal injury. OTA eventually causes inflammation, glomerular and tubular damage, and even renal fibrosis through a different molecular signaling pathway. A summary mechanism of recent evidence of OTA-induced nephrotoxicity was in Figure 1. Reducing ROS production, activation of the Nrf2 pathway, maintaining DNA stability, or using nanoparticles as mycotoxin adsorbents are several ways used to prevent OTA toxicity. Both humans and animals will inevitably face exposure to OTA-contaminated food; the disclosed molecular mechanism of OTA nephrotoxicity is an important issue which provided further opinions and strategies such as prophylactic food or feed additive which could counteract OTA toxicity to protect human and animal health.

## Figures and Tables

**Figure 1 ijms-22-11237-f001:**
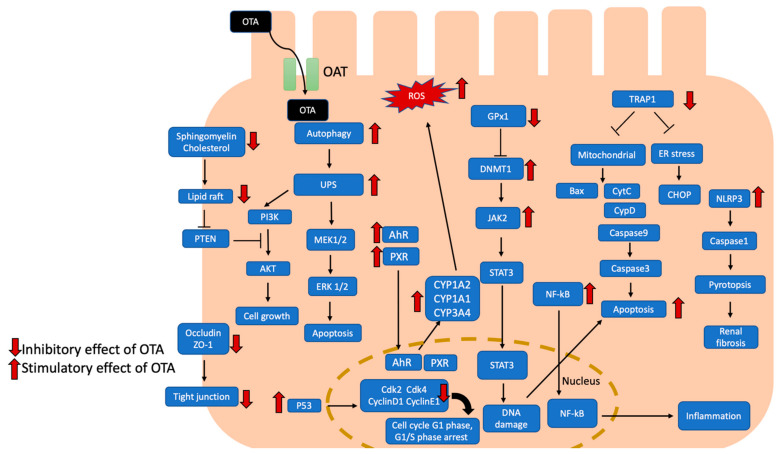
Effects of OTA-induced nephrotoxicity are illustrated, and the potential signal pathways are summarized. OTA uptake by proximal tubular cells with organic anion transporter. OTA activates AhR and PXR signaling pathways to induce cytochrome P450 1A1 (CYP1A1), CYP1A2, and CYP3A4, leading to increased ROS production. Meanwhile, OTA induced autophagy and UPS led to apoptosis with MEK1/2/ERK1/2 pathway. OTA attenuated cholesterol and sphingomyelin to disrupt lipid raft formation which led to cell apoptosis through PTEN/PI3K/AKT pathway. Besides, OTA inhibited occluding and ZO1 to disrupt tight junction. In addition, OTA activated DNMT1-dependent JAK2/STAT3 signaling pathway to induce apoptosis and DNA damage. OTA could promote NF-kB translocated to nucleus which induced inflammation. OTA also inhibits TRAP-1 which regulated mitochondria-mediated and ER stress-excited apoptosis. Furthermore, OTA activated NRPL3 inflammasome and caused caspase1-dependent pyroptosis. OTA also activated p53 which suppressed cyclin D1, CDk2 and CDk4, leading to cell cycle arrest. AhR, aryl hydrocarbon receptor; Akt, serine/threonine-specific protein kinases; Bax, Bcl-2 associated x protein; Cdk2, cyclin dependent kinase 2; Cdk4, cyclin dependent kinase 4; CHOP, CAAT/enhancer-binding protein (C/EBP) homologous protein; CYP1A1, cytochrome P450 1A1; CYP1A2, cytochrome P450 1A2; CYP3A4, cytochrome P450 3A4; CytC, cytochrome C; DNMT1, DNA methyltransferase 1; ER stress, endoplasmic reticulum stress; ERK 1/2, extracellular signal-regulated kinases 1/2; JAK2, Janus kinase 2; MEK 1/2, mitogen-activated protein kinase kinase 1/2; NF-κB, nuclear factor kappa-light-chain-enhancer of activated B cell; NLRP3, NLR family pyrin domain containing 3; OAT, organic anion transporter; PTEN, phosphatase and tensin homolog; PI3K, phosphoinositide 3-kinases; PXR, pregnane X receptor; ROS, reactive oxygen species; STAT3, signal transducer and activator of transcription 3; TRAP-1, tumor necrosis factor receptor-associated protein 1; UPS, ubiquitin-proteasome system; ZO-1, Zonula occludens protein 1.

**Table 1 ijms-22-11237-t001:** In vitro OTA-induced nephrotoxicity studies.

Cell Type	Time	Dose	Result/Conclusion	Reference
HK-2	24 h	2–8 uM	OTA induced HK-2 apoptosis by regulating PTEN/Akt signaling pathway through impaired lipid raft formation	Song et al. [40]
HK-2	48 h	200 nM	OTA induced nephrotixicity through AhR Smad2/3 HIF-1α signaling pathway	Pyo et al. [41]
HK-2	24 h	0.2, 1, 5 uM	Oleanolic acid (OA) promoted TRAP-1 to relieve mitochondrial mediated and ER stress excited apoptosis	Zhang et al. [42]
HK-2	1, 3, 6, 12, 24 h	10 uM	OTA activated NF-kB to induce ERK 1/2-dependent apoptosis	Darbuka et al. [43]
PTEC	24, 72 h	0–100 uM	OTA downregulated transciptional expression of GST leading to nephrotoxicity; organic anionic membrane transporter(s) are involved in the excretion of OTA	Imoaka et al. [89]
HKC	24 h	20 uM	OTA cause G0/G1 phase arrest; OTA altered methylation state of specific gene which regulated cell cycle	Zhang et al. [58]
HK-2	24 h	10, 25 uM	OTA suppressed cyclin D1, CDk2, and CDk4 through p53	Asci et al. [65]
HEK 293	24 and 48 h	0.125–0.5 uM	OTA promoted expression of PDK1, HIF-1α, TGF-β, VEGF and EPO; OTA increased ATP production	Raghubeer et al. [87]
HK-2, MEF	1–24 h	10 uM	OTA induced autophagy and UPS to activate PI3K/AKT and MAPK/ERK1–2 signaling pathway	Akpinar et al. [77]
PK 15, PAM	24 h	2–8 ug/mL	OTA induced cytotoxicity, apoptosis, DNA damage through DNMT1-JAK2/STAT3 -SOCS3 signaling pathway	Gan et al. [53]
PK 15	48 h	2–8 uM	Protective autophagy induced by OTA through inhibition of ATK/mTOR signaling pathway	Qian et al. [78]
MDCK	24 h	0–1.2 ug/mL	OTA reduced occludin and ZO-1; alpha-tocopherol maintain occludin, ZO-1	Fusi et al. [81]
HK-2	24 h	0–50 uM	OTA induced H3K9 hypoacetylation, leading to suppressed gene expression	Limbeck et al. [57]
HKC, NRK-52E	72 h	1–2 uM	OTA induced senescence by activating p53-p21 and p16-pRB signaling pathway	Yang et al. [84]
PK 15, porcine primary splenocyte	24 h	0.5–8 ug/mL	OTA induced nephrotoxicity through p38 pathway, OTA induced immunotoxicity through ERK pathway	Gan et al. [44]
HK-2, HEK293T	24 h	10, 100 nM	OTA downregulated CDK2 leading to G1 phase cell cycle arrest	Dubourg et al. [66]

HK-2, Human proximal tubular epithelial cells; HKC, human renal proximal tubular cells; HEK 293, HEK293T human embryonic kidney cells; MDCK, Madin-Darby canine kidney cells; MEF, mouse embryonic fibroblast; NRK-52E, rat renal tubular duct epithelial cells; PTEC, renal proximal tubular epithelial cells; PK, 15 porcine kidney epithelial cells; PAM, porcine alveolar macrophage cells.

**Table 2 ijms-22-11237-t002:** In vivo OTA-induced nephrotoxicity studies.

Animal Model	Cell Type	Time	Dose	Result	Reference
C57BL/6 mice OTA: 1.0, 2.0 mg/kg IP; 14 days	MDCK	24 h	0–4 uM	OTA activated NLRP3 inflammasome and caspase-1 dependent pyroptosis	Li et al. [46]
C57BL/6 mice OTA 0.5, 1.5, 2.5 mg/kg IP; 3 weeks	HMC	48 h	0–8 uM	OTA induced glomerular injury by ERK/NF-kB pathway in vivo and in vitro	Le et al. [80]
ICR mice OTA: 200, 1000 ug/kg; 12 weeks	HK-2	48 h	50–200 nM	OTA induced EMT and renal fibrosis by TGF-β/Smad2/3 and B-catenin/Wnt signaling pathway in vivo and in vitro	Pyo et al. [79]
Piglet OTA: 50, 200 ug/kg; 28 days				OTA upregulated expression of miR-497, miR-133a-3p, miR-423-3p, miR-34a, miR-542-3p and downregulated expression of miR-421-3p, miR-490, and miR-9840-3p	Marin et al. [74]
SPF F344, Wistar rat OTA: 0, 70, 210 ug/kg; 13 weeks	HKC	48 h	0–25 uM	OTA induced GRP75 to prevent renal injury and mitochondrial dysfunction	Yang et al. [73]
ICR mice OTA: 1–3 ug/kg; 6 weeks	HK-2	48 h	25–200 nM	OTA induced ROS production through AhR, PXR and Nrf2 signaling pathway	Lee et al. [31]
Rats OTA: 70, 210 ug/kg; 4 weeks, 13 weeks	NRK-52E	24 h	20, 50 uM	OTC2 modulated OTA-induced apoptosis	Qi et al. [45]
HO-1 knock out mice OTA: 2.5 mg/kg; IP; 20 days	LLC-PK1	20–60 min, 24 h	25 uM	HO-1 mitigated OTA nephrotoxicity thorough regulating Nrf-2, miR-34a, and miR-21	Loboda et al. [86]
Nrf-2 knock out mice OTA: 2.5 mg/kg; IP; 20 days	LLC-PK1	24 h	25 uM	OTA induced nephrotixcity is sex-dependent: increased renal injury in male Nrf-2 knockout mice	Loboda et al. [88]

HK-2, Human proximal tubular epithelial cells; HKC, human renal proximal tubular cells; HMC, human mesangial cells; ICR mice, institute of cancer research mice; LLC-PK1, pig kidney epithelial cells; MDCK, Madin-Darby canine kidney cells; NRK-52E, rat renal tubular duct epithelial cell line; SPF F344 rats, specific pathogen-free Fischer 344 rats.

**Table 3 ijms-22-11237-t003:** Prevention of OTA-induced nephrotoxicity studies.

Protective Agent	Animal Model	Cell Type	OTA Dose	Result/Conclusion	Reference
Oleanolic acid (OA)		HK-2	0.2, 1, 5 uM	OA promoted TRAP-1 to relieve mitochondrial mediated and ER stress excited apoptosis	Zhang et al. [42]
Curcumin (CURC)	SD rats; OTA 0.5 mg/kg; 14 days			Curcumins attenuated OTA-induced nitosative stress, inflammatory, and DNA damage in kidney and liver of rats	Longobardi et al. [34]
Taurine		PK 15	1–6 uM	Taurine reversed apoptosis, increasing LDH level induced by OTA	Liu et al. [92]
Luteolin (LUT)		NRK-52E	50 uM	LUT alleviated ROS production, loss of mitochondrial membrane potential induced by OTA; LUT enhanced expression of Nrf2 and HIF-1α	Liu et al. [114]
N-Acetyl-L-Tryptophan (NAT)		HEK 293T	4 ug/mL	NAT ameliorated OTA-induced cell cycle arrest, mitochondrial membrane potential disturbance, protein inhibition	Argawa et al. [61]
Astaxanthin(ASX)	C57BL/J mice OTA 5 mg/kg; 27 days			ASX ameliorated apoptosis, oxidative stress induced by OTA; ASX activate Nrf2/Keap1 signaling pathway	Li et al. [100]
Curcumin (CURC)	SD rat OTA 0.5 mg/kg: 14 days			Curcumins maintained GFR, attenuated oxidative stress, glomerular, and tubular damage, tubular interstitial fibrosis	Damino et al. [95]
Hydroxytyrosol (HT)	SD rat OTA 250 ug/kg: 90 days	MDCK, LLC-PK1, RK 13	2.5 ug/mL	HT decreased ROS production, enhanced cell viability; HT decreased renal fibrosis, oxidative stress in vitro after OTA exposure.	Crupi et al. [33]
Selenium Yeast (Se-Y)	chicken OTA 50 ug/kg, 80 days			Se-Y against OTA induced apoptosis, oxidative stress, renal injury in chicken	Li et al. [115]
Copper nanoparticles and aluminum silicate nanoparticles	Nile tilapia fish OTA 1 mg/kg; 6 weeks			Nanoparticle aluminum silicate or copper ameliorated OTA-induced liver and kidney injury	Fadl et al. [111]
Selenium		PK 15	4 ug/mL	Selenium promoted GPx1 expression to decrease DNMT1, DNA damage after OTA exposure.	Gan et al. [103]
Yemeni green coffee powder	Wistar rats OTA 10 mg/kg; 28 days			Yemeni green coffee restored SOD, glutathione level; it reduced kidney injury after OTA exposure	Nogaim et al. [96]
Red orange and lemon extract (RLE)	SD rats OTA 0.5 mg/kg/day; 14 days			RLE reversed GSH level and GFR; it reduced oxidative stress, renal fibrosis, glomerual damage and tubular damage	Damino et al. [97]
Ursolic acid (UA)		HEK 293T	8 uM	UA decreased ROS production, enhanced cell viability, reversed inhibition of LonP1 by OTA	Li et al. [90]
Troxerutin	Specific-pathogen–free male CD1 (ICR) mice OTA 1 mg/kg; 4 weeks, 12 weeks			Troxerutin alleviated OTA-induced lipid accumulation, lipid peroxidation, increased TG and SMase level	Yang et al. [67]
Zinc supplement		MDCK	1.0 ug/mL	Zinc supplement suppressed ROS production and apoptosis through enhanced metallothionein	Li et al. [108]
δ-tocotrienol (Delta)	male, SD rats OTA, 0.5 mg/kg, gavage, 14 days			δ-tocotrienol alleviated ROS production induced by OTA	Damino et al. [93]
rMnSOD	male, SD rats OTA, 0.5mg/kg, gavage, 14 days			rMnSOD restored suppression of fluid reabsorption, decreased NHE3 and NO production by OTA	Damino et al. [35]
Sulforaphane (SFN)	Nrf-2 knock out mice: 2.5 mg/kg; IP; 20 days	LLC-PK1	25 uM	SFN alleviated OTA-induced inflammatory cytokine and apoptotic factor	Laboda et al. [88]
Selenoprotein S (SelS)		PK 15	1–4 ug/mL	SelS alleviated OTA-induced apoptosis and ROS production	Gan et al. [105]
Selenium probiotics (SP)	Piglet, OTA 0.4 mg/kg; 6 weeks			SP restored GPx, SOD against OTA-induced kidney injury	Gan et al. [106]
Chitosan nanoparticles (COS) plus quercetin (Q)	SD rats 3 mg/kg; diet; 3 weeks			COS alone or plus Q mitigated OTA-induced oxidative stress and apoptosis	Abdel et al. [113]

HEK293T human embryonic kidney cells; HK-2, Human proximal tubular epithelial cells; ICR mice, institute of cancer research mice; LLC-PK1, pig kidney epithelial cells; MDCK, Madin-Darby canine kidney cells; NRK-52E, rat renal tubular duct epithelial cell line; PK 15 porcine kidney epithelial cells; RK 13, rabbit kidney cell line; SD rats, Sprague-Dawley rat.

## Data Availability

Not applicable.

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
