# Peer review of "Ochratoxin A-Induced Nephrotoxicity: Up-to-Date Evidence"

_ijms, 2021, doi:10.3390/ijms222011237_

Round 1

Reviewer 1 Report

The presented manuscript is interesting, however it can't be accepted in present form.

General: 

I am not native English speaker, however in my opinion the manuscript requires native English editing.

If you use abbreviation for the first time, please explain it in text. Not all abbreviations are collected in the "Abbreviations" section of the manuscript.

Specific:

Line 30-31: please specify the deleterious effect on human health.

Line 49-50: how OTA is excreted mostly?

Line 104, line 170: HK-2 cells

Line 127-129: I don't understand the statement.

Line 180-193: this whole paragraph is a mess. First you describe mitochondrial disfuction and ATP level and out of blue comes Hsp. Please connect those information.

Line 201-202: what is the conclusion?

Line 218: where the inflammation is induced by NF-kB by OTA? In kidneys or other organ?

Line 232: what sex is prone to nephrotoxicity? 

Line 254: I would rearrange the whole paragraph. I would put antioxidants together.

Line 303: please cite some papers about luteolin and its antioxidant properties.

Line 306-309: Astaxanthin belongs to keto-carotenoid group not to flovonoid like luteolin, so I don't understand why you put it together.

Line 320: oxidative damage and apoptosis in a chicken? I don't understand. Whole body or organs? If so, which organs?

Line 331: "Zinc can akso regulate..." please cite some paper.

Line 348-358: please rewrite the conclusion. In present state this conclusion weak.

Author Response

Response to reviewers’ comments:

Dear Editor-in-chief:

Many thanks for the hard work and numerous useful suggestions you provided on our manuscript. We followed the reviewers’ suggestions, and the detailed corrections are listed below, point-by-point, based on the issues have been raised.

Response to Reviewer 1’s Comments:

  • If you use abbreviation for the first time, please explain it in text. Not all abbreviations are collected in the "Abbreviations" section of the manuscript.

Ans: We wrote the full name before using abbreviations. We rechecked the all abbreviations are collected in the “abbreviations”.

Specific:

  • Line 30-31: please specify the deleterious effect on human health

Ans: We re-wrote the sentence “OTA has nephrotoxic deleterious effect on human health due to exposure to OTA-contaminated food” and reattached ref [5] “Ochratoxin A in human kidney diseases. Food Addit Contam 2005, 22 Suppl 1, 53-7” (line 30-31, page 1)

  • Line 49-50: how OTA is excreted mostly?

Ans: After literature review, OTA could be excreted by kidney, feces, and breast milk; but it didn’t mention which way is excreted mostly of OTA. References: “Toxicokinetics and toxicodynamics of ochratoxin A, an update. Chem Biol Interact 2006, 159, (1), 18-46”; ” Ochratoxin A: Molecular Interactions, Mechanisms of Toxicity and Prevention at the Molecular Level, Toxins (Basel). 2016 Apr 15;8(4):111. “

  • Line 104, line 170: HK-2 cells

Ans: We wrote the full name “human proximal tubular epithelial cell (HK-2)” for the first before used abbreviation. (line 115-116, page 3)

  • Line 127-129: I don't understand the statement.

Ans: We re-wrote the statement “OTA induced delete mutation of homologous repair of DNA double-strand break led to genotoxicity, p53 could suppress delete mutation of homologous repair after OTA exposusre.”. (line 150-152, page 3)

  • Line 180-193: this whole paragraph is a mess. First you describe mitochondrial disfuction and ATP level and out of blue comes Hsp. Please connect that information.

Ans: We rewrote the statement as “Mitochondrial membrane potential play a essential role in mitochondrial homeostasis. Transmembrane potential of hydrogen ion was formed by mitochondrial membrane potential is used to make ATP. Rising or decrease of mitochondrial membrane potential will affect cell viability[69]. Argawal et al. found loss of mitochondrial membrane potential at the initiation of OTA treatment (4-12 hrs), followed by hyperpolarization of mitochondrial membrane potential at 24hrs [62]. Eder et al discovered that OTA induced hyperpolarization of mitochondrial membrane potential and increased cellular ATP [70]. In contrast, Chebotareva et al. found OTA decreased intracellular ATP, which can be reversed by Diosmetin (DIOS)[71]. In addition, OTA inhibited mitochondrial oxygen consumption through suppressed complex I and II of the mitochondrial respiratory chain, which implying that OTA caused mitochondrial dysfunction in rat renal proximal tubule [72]. Heat shock protein (HSP) play an intracellular defensive role in the kidney, which promote the ability to against apoptosis and necrosis [73], glucose-regulated protein 75 (GRP-75) belongs to the HSP70 family, it may protect the renal cell from mitochondrial dysfunction through regulating Lonp1 after OTA exposure. Besides, decrease expression of GRP75 was with increasing OTA dose, thus GRP-75 could be a biomarker of renal tubular necrosis induced by foodborne toxicity like OTA [74].” And added a new reference [69] “Mitochondrial membrane potential. Anal Biochem 2018, 552, 50-59”. (line 238-254, page 5)

  • Line 201-202: what is the conclusion?

Ans: We rewrote the sentence and added the conclusion as” Stachurska et al. found that OTA increased miR-132, miR-200 by ROS, and TGF-β to inhibit Nrf-2, HO-1 which led to suppressed proliferation and cell viability in renal proximal tubular epithelial cells [77]” (line 262-264, page 5)

  • Line 218: where the inflammation is induced by NF-kB by OTA? In kidneys or other organ?

Ans: We rewrote the sentence “OTA induced inflammation of human embryonic kidney cells (HEK293) through the NF-kB pathway” (Line 378-379, page 6)

  • Line 232: what sex is prone to nephrotoxicity? 

Ans: Male is prone to nephrotoxicity; we rewrote the sentence “Loboda et al. found that OTA-induced nephrotoxicity is sex-dependent which male is more susceptible and regulated by the Nrf-2 pathway.” (line 400-402, page 6)

  • Line 254: I would rearrange the whole paragraph. I would put antioxidants together.

Ans:  We rearrange the whole paragraph and added subtitle as 3.1 antioxidant, 3.2 Nrf2 activator, 3.3 Trace element, 3.4 Nanoparticle from (line 433, page 9 to line 648, page 12)

  • Line 303: please cite some papers about luteolin and its antioxidant properties.

Ans: We attached referrence [102] “Luteolin alleviates inorganic mercury-induced kidney injury via activation of the AMPK/mTOR autophagy pathway. J Inorg Biochem 2021, 224, 111583”and referrence [103] “Luteolin protects against lead acetate-induced nephrotoxicity through antioxidant, anti-inflammatory, anti-apoptotic, and Nrf2/HO-1 signaling pathways. Mol Biol Rep 2020, 47, (4), 2591-2603.” We rewrote the paragraph as “LUT is a dietary flavonoid that has antioxidant properties from recent reports like LUT restored autophagy though AMPK pathway to mitigate mercury induced destruction of renal tubule, inflammation, oxidative stress, apoptosis of rats kidney[102]. LUT alleviated lead induced kidney injury by inhibit oxidative stress, inflammation, apoptosis and enhanced expression of Nrf2, HO-1 in vitro study[103]. Liu et al. have found that cotreatment of LUT promoted expression of Nrf2 and antioxidant enzyme to ameliorate ROS level and mitochondrial membrane potential due to OTA exposure in rat kidney cells. In addition, LUT activated HIF-a to regulated angiogenesis and epithelial restituition process. These results suggest LUT is a potential dietary flavonoid to counteract other myocotoxin or environmental toxins [104].”(line 565-574, page 11)

  • Line 306-309: Astaxanthin belongs to keto-carotenoid group not to flovonoid like luteolin, so I don't understand why you put it together.

Ans: We put Astaxanthin in another paragraph (line 509-514, page 10)

  • Line 320: oxidative damage and apoptosis in a chicken? I don't understand. Whole body or organs? If so, which organs?

Ans: Oxidative damage and apoptosis in kidney of chicken. We rewrote the sentence as “Cotreatment of selenium yeast (Se-Y) promoted PI3/AKT and Nrf2/Keap1 signaling pathways to alleviate OTA-induced oxidative damage and apoptosis in kidney of chickens.” (line 585-587, page 11)

  • Line 331: "Zinc can also regulate..." please cite some paper.

Ans: We added reference [113] “Zinc protects HepG2 cells against the oxidative damage and DNA damage induced by ochratoxin A. Toxicol Appl Pharmacol 2013, 268, (2), 123-31”and reference [114]” Zinc-dependent changes in oxidative and endoplasmic reticulum stress during cardiomyocyte hypoxia/reoxygenation. Biol Chem 2020, 401, (11), 1257-1271.” We rewrote the sentence as “Li et al. found that pretreatment of zinc supplements promotes metallothionein (MT) expression to reduce ROS production and apoptosis in kidney epithelial cells after OTA exposure [112]. Besides, zinc supplement reduced ROS production and increase expression of SOD, furthermore, zinc supplement maintained DNA stability by reducing DNA strand break, 8-OHdG formation and DNA hypomethylation after OTA exposure in HepG2 cells[113]. And also, Bodiga disclosed zinc replenishment suppressed intracellular superoxide production, apoptosis, attenuated ER stress in hypoxia/reoxygenation of cardiomyocytes[114]. Same as selenium, Zinc acts as ROS scavenger, maintain DNA stability and it has antiapoptotic property to against OTA induced toxicity.” (line 605-610, page 11; line 631-634, page 12)

  • Line 348-358: please rewrite the conclusion. In present state this conclusion weak.

Ans: We rewrote the conclusion as “The mechanisms of OTA-induced nephrotoxicity are complicated and uncomplete clarified; the literature review found that OTA activates PTEN/AKT, ERK1/2, NF-kB, p38 pathways to induced apoptosis. Meanwhile, OTA induced lipotoxicity, pyroptosis, and senescence of renal cells. Besides, OTA regulates transcriptional factors, DNA methylation, and histone medication to cause renal injury. OTA eventually cause inflammation, glomerular and tubular damage, even renal fibrosis through different molecular signaling pathway. A summary mechanism of recent evidence of OTA-induced nephrotoxicity was in figure 1. Reducing ROS production, activation of Nrf2 pathway, maintaining DNA stability, or using nanoparticles as mycotoxin adsorbents are several ways used to prevent OTA toxicity. Both humans and animals will inevitably exposure of OTA contaminated food, disclosed molecular mechanism of OTA nephrotoxicity is an important issue which provided further opinions and strategies like prophylactic food or feed additive could counteract OTA toxicity to protect human and animal health.” (line 659-671, page 14)

Reviewer 2 Report

Ochratoxin A (OTA) is a known and important problem in food/feed, largely due to its unique chemistry and toxicity (renal accumulation). Therefore it is important to assess existing mechanisms related to OTA's nephrotoxicity. With that said the authors effort is worthwhile. However the review is very descriptive and the authors summarize published in vivo/in vitro studies in an unorganized manner. Basically, there are no connections between Section 2 and Section 3. Furthermore, among all the mechanisms listed in section 2, there are no connections among them. Are these mechanisms/pathways work independently, additively, or synergically? Same holds for section 3. Other important components missing in the review include but not limited to any existing challenges that need to be addressed? Future trends/directions? Pro/cons and limitations of published in vivo/in vitro studies. As the authors pointed out in the review the mechanisms of OTA induced nephrotoxicity are complicated so it requires more efforts to comb, analyze and present scientific evidences. Otherwise the review would not help readers to get a better understand about the topic.      

Author Response

Response to reviewers’ comments:

Dear Editor-in-chief:

Many thanks for the hard work and numerous useful suggestions you provided on our manuscript. We followed the reviewers’ suggestions, and the detailed corrections are listed below, point-by-point, based on the issues have been raised.

Response to Reviewer 2’s Comments:

  • Basically, there are no connections between Section 2 and Section 3. Furthermore, among all the mechanisms listed in section 2, there are no connections among them. Are these mechanisms/pathways work independently, additively, or synergically? Same holds for section 3.

Ans: From Section 2 and Section 3, most of these mechanism/pathways work independently.

But the following study are belong to same pathway.

  1. Crupi et al. showed OTA increases NO production in different kinds of kidney cell lines [33]. Hydroxytyrosol is a phenolic compound with antioxidant, antimicrobial, antitumoral, cardioprotective, and neuroprotective properties. Pretreatement of hyroxytyrosol prevented OTA-induced renal injury by reducing ROS level, nitrite production, kidney fibrosis in vivo and in vitro. This study provided a evidence that phenolic compound could against oxidative stress related renal injury [33]. Ref 33” Protective Effect of Hydroxytyrosol Against Oxidative Stress Induced by the Ochratoxin in Kidney Cells: in vitro and in vivo Study. Front Vet Sci 2020, 7, 136”
  2. Longobardi et al. reveal that OTA promoted iNOS expression and increased NO production in vitro study [34]. Longobardi et al. demonstrated that cotreatment of curcumin attenuated OTA-induced nitrosative stress and inflammatory and DNA damage by reduced NO production, iNOS expression, TNF-α, IL-1B, IL-6, and 8-Hydroxy-2′-Deoxy Guanosine (8-OHdG) in kidney and liver of rats [34]. Ref 34 ” Curcumin Modulates Nitrosative Stress, Inflammation, and DNA Damage and Protects against Ochratoxin A-Induced Hepatotoxicity and Nephrotoxicity in Rats. Antioxidants (Basel) 2021, 10,1239.”
  3. In contrast, Damiano et al. found OTA reduced NO production in rat kidneys which may be related to OTA-induced hypertension by inhibiting NO synthase (NOS). Besides, OTA enhanced superoxide production which inhibit sodium-hydrogen exchanger isoform 3 (NHE3) lead to disturbance of fluid reabsorbtion. Therefore, alteration of proximal tubular reabsorption by OTA induced ROS would affect balance between fluid and solute [35]. In addition, cotreatment of rMnSOD increases the production of nitric oxide(NO), restore impaired sodium-hydrogen exchanger isoform 3 (NHE3) regulating fluid reabsorption in the proximal tubule, decreases blood pressure after OTA exposure in rat kidney[35]. Ref 35. “Effect of rMnSOD on Sodium Reabsorption in Renal Proximal Tubule in Ochratoxin A-Treated Rats. J Cell Biochem 2018, 119, (1), 424-430”
  4. Argawal et al. using molecular docking analysis revealed that OTA could bind on a common pocket of phenylalanine-tRNA synthase to disrupt protein synthesis [62].Argawal et al. found loss of mitochondrial membrane potential at the initiation of OTA treatment (4-12 hrs), followed by hyperpolarization of mitochondrial membrane potential at 24hrs [62]. Pretreatment of N-Acetyl-L-Tryptophan (NAT) could bind on phenylalanine tRNA synthetase may decrease OTA-induced protein inhibition; in addition, NAT also protected HEK cells from ROS insult, cell cycle arrest, disturbed mitochondrial membrane potential induced by OTA. Prophylatic dietary application of NAT may prevent OTA induced toxicity[62]. Ref 62. “Amelioration of ochratoxin-A induced cytotoxicity by prophylactic treatment of N-Acetyl-L-Tryptophan in human embryonic kidney cells. Toxicology 2020, 429, 152324.”
  5. As evidenced by lipid deposition, OTA-induced renal lipotoxicity was found in the renal cortex and medulla after OTA exposure. In addition, OTA induced triglyceride levels elevation in the kidney. Moreover, OTA suppressed the expression of PPAR-α and medium-chain acyl-CoA dehydrogenase (Mcad), which is needed for fatty acid oxidation. Meanwhile, OTA promoted lipid peroxidation and sphingomyelinase, which might be involved in apoptosis. Besides, OTA inhibited activity of brown adipose tissue (BAT) to disturb energy metabolism balance [68]. Troxerutin is a natural flavonoid from rutin that has anti-inflammation and antioxidant properties. Cotreatment of troxerutin mitigate lipid accumulation in renal cells, relieved lipid peroxidation, reversed level of PPAR-α, medium-chain acyl-CoA dehydrogenase (Mcad) and sphingomyelinase (SMase) to alleviate OTA induced renal lipotoxicity. In addition, Troxerutin improved whole body energy metabolism by attenuated OTA suppressed brown adipose tissue (BAT) activity[68]. Ref 68. “Precision toxicology shows that troxerutin alleviates ochratoxin A-induced renal lipotoxicity. Faseb j 2019, 33, (2), 2212-2227.”

  • Other important components missing in the review include but not limited to any existing challenges that need to be addressed? Future trends/directions? Pro/cons and limitations of published in vivo/in vitro

Ans: We review the literature and added the following sentence to clarify future trends, directions, Pro/cons and limitation of these published in vivo/in vitro studies

  1. “Therefore, modulation of AhR, PXR and Nrf2 could be considered to prevent or treatment of OTA induced toxicity [31].” (line 99-100, page 2)
  2. “Besides, OTA enhanced superoxide production which inhibit sodium-hydrogen exchanger isoform 3 (NHE3) lead to disturbance of fluid reabsorbtion. Therefore, alteration of proximal tubular reabsorption by OTA induced ROS would affect balance between fluid and solute [35].” (line 108-111, page 3)
  3. “Transcriptomic analysis found that hypoxia, epithelial-to-mesenchymal transition (EMT) , apoptosis and xenobiotic metabolism pathway are affected by OTA. The following result showed OTA-induced EMT, apoptosis, and kidney injury by regulating AhR-Smad3/HIF-1α signaling pathway. This study provided an opinion to against OTA induced renal toxicity through understanding of OTA induced EMT and apoptosis related molecular pathway [42].” (line 126-131, page 3)
  4. “Zhang et al. discovered tumor necrosis factor receptor-associated protein 1 (TRAP-1) enhanced expression of Bcl-2 and inhibited expression of CypD, Bax, GRP78, CHOP to against apoptosis. OTA could inhibit TRAP-1 led to mitochondria-mediated and ER stress-excited apoptosis of proximal tubular cells [43].” (line 131-134, page 3)
  5. “Another study found OTA induced nephrotoxicity and apoptosis through the p38 pathway in in porcine kidney epithelial cells (PK 15); meanwhile, OTA induced immunotoxicity through ERK signaling pathway in porcine primary splenocyte. This study clarified that, OTA induced toxicity through different molecular pathway in different cell [45].” (line 136-140, page 3)
  6. “Qi et al. discovered organic cation transporters 2 (OCT2) participate in OTA-induced cell apoptosis in rat proximal tubule cells (NRK-52E) cells by increase caspase 3 and cyclin dependent kinase 1 (CDK1), but cell transport experiment would be conducted to ascertain the role of OCT2 in OTA induced toxicity [46]. Besides apoptosis, OTA induced nephrotoxicity via activating NOD like receptor protein 3 (NLRP3) inflammasome and caspase1-dependent pyroptosis lead to renal fibrosis by evidence of increasing TGF-β and α-SMA in vivo and in vitro study. This study offer a novel opinion or strategy to treatment of OTA induced renal fibrosis [47].” (line 140-147, page 3)
  7. “OTA induced delete mutation of homologous repair of DNA double-strand break lead to genotoxicity, p53 could suppress delete mutation of homologous repair after OTA exposusre. In addition, p53 attenuated OTA induced karyomegaly and apoptosis in mice kidneys [50].” (line 150-152, page 3; line 162, page 4)
  8. “OTA activated DNA methytransferanse1 (DNMT1) dependent JAK2/STAT3 signaling pathway to induce apoptosis and DNA damage in PK15 cells and porcine alveolar macrophage (PAM); meanwhile, OTA increased the expression of suppressors of cytokine signaling3 (SOCS3), which was regulated by DNMT1 only in PK15 cells. This study elucidated that OTA induced nephrotoxicity instead of immunotoxicity through DNMT1/JAK2/STAT3/SOCS3 signaling pathway[54].” (line 171-177, page 4)
  9. “This study disclosed loss of histone acetyltransferase is a critical step to OTA induced carcinogensis, but alteration of non-histone acetylation by proteome-wide analysis should be conducted to clarify the relationship between HAT inhibition and OTA mediated gene expression change, mitosis and DNA repair[58]. In addition, OTA induced cell cycle arrest in G0/G1 phase of HK-2. Some gene was related to G0/G1 cell cycle arrest, but only increase expression of PTEN due to OTA-induced hypomethylation of PTEN which involved in G1/S phase transition. G0/G1 phase cell cycle arrest, inhibition of Notch and Ras/MAPK/CREB signaling pathway after OTA exposure would lead to renal toxicity[59].” (line 188-196, page 4)
  10. “Recent studies discovered OTA-induced cell cycle arrest in the G1 and G1/S phase mediated by p53 through suppression of cyclin D1, Cdk2 and Cdk4 in HK-2. This study suggested the status of gene and protein should be showed after regulated by upper mediator like p53, because many gene, proteins, intermediate molecular even upper mediator could regulate cell cycle [66]. Using weight correlation network analysis by Duborg et al. found that Cyclin-dependent kinase 2 (CDK2) plays a major regulator in G1 cycle arrest induced by OTA. OTA inhibit expression of CDK2, overexpression of CDK2 attenuated OTA induced G1 phase cycle arrest. [67].” (line 211-212, page 4; line 223-228, page 5)
  11. “Besides, OTA inhibited activity of brown adipose tissue (BAT) to disturb energy metabolism balance [68]”. (line 235-236, page 5)
  12. We rewrote the whole paragraph as “Mitochondrial membrane potential play a essential role in mitochondrial homeostasis. Transmembrane potential of hydrogen ion was formed by mitochondrial membrane potential is used to make ATP. Rising or decrease of mitochondrial membrane potential will affect cell viability[69]. Argawal et al. found loss of mitochondrial membrane potential at the initiation of OTA treatment (4-12 hrs), followed by hyperpolarization of mitochondrial membrane potential at 24hrs [62]. Eder et al discovered that OTA induced hyperpolarization of mitochondrial membrane potential and increased cellular ATP [70]. In contrast, Chebotareva et al. found OTA decreased intracellular ATP, which can be reversed by Diosmetin (DIOS)[71]. In addition, OTA inhibited mitochondrial oxygen consumption through suppressed complex I and II of the mitochondrial respiratory chain, which implying that OTA caused mitochondrial dysfunction in rat renal proximal tubule [72]. Heat shock protein (HSP) play an intracellular defensive role in the kidney, which promote the ability to against apoptosis and necrosis [73], glucose-regulated protein 75 (GRP-75) belongs to the HSP70 family, it may protect the renal cell from mitochondrial dysfunction through regulating Lon protease 1 (Lonp1) after OTA exposure. Besides, decrease expression of GRP75 was with increasing OTA dose, thus GRP-75 could be a biomarker of renal tubular necrosis induced by foodborne toxicity like OTA [74].” (line 238-254, page 5)
  13. “Stachurska et al. found that OTA increased miR-132, miR-200 by ROS, and TGF-β to inhibit Nrf-2, HO-1 which lead to suppressed proliferation and cell viability in renal proximal tubular epithelial cells [77].” (line 262-264, page 5)
  14. “Meanwhile, OTA increased PI3K/AKT and MAPK/ERK1-2 signaling pathways by degradation of phosphatases dual specificity phosphatase 3 (DUSP3), dual specificity phosphatase 4 (DUSP4) and Ph domain and Leucine rich repeat protein phosphatase (PHLPP) through autophagy and UPS which may involved in OTA toxicity and carcinogenicity [78]. However, Qian et al. found autophagy could protect PK cells from OTA-induced apoptosis by suppressing Akt/ mTOR signaling pathway. This study provided a novel therapeutic agent to against OTA toxicity through autophagy [79].” (line 269-271, page 5; line 364-367, page 6)
  15. “Epithelial-to-mesenchymal transition (EMT) is involved in renal fibrosis, and an experiment discovered OTA activated TGF-β/Smad2/3 and B-catenin/Wnt signaling pathways to induce EMT and renal fibrosis by evidence of increasing expression of α-SMA and fibronectin. It is possible to prevent of OTA induced EMT related renal fibrosis through regulating TGF-β/Smad2/3 and B-catenin/Wnt pathways[80].” (line 369-373, page 6)
  16. “OTA induced senescence may provide a condition to become cancer cell, therefore understanding of senescence mechanism would help to study OTA induced carcinogenesis [85].” (line 385-387, page 6)
  17. “In addition, this study suggest further experiment could be conducted to clarify the role of HO-1 in OTA induced genotoxicity [87].” (line 391-392, page 6)
  18. “Exposure to OTA may increase angiogenesis by increasing the expression of HIF-1α and EPO gene after 24 hours; meanwhile, expression of TGF-β and vascular endothelila growth factor (VEGF) increase significantly at 48 hours. OTA may interrupt the normal metabolic process of thorough alteration of ATP production and pyruvate dehydrogenase 1(PDK1) expression. HIF-1α regulate cellular adaptation to hypoxia and hypoxic condition may promote carcinogenesis, therefore investigate OTA induced hypoxic condition may help to clarify OTA related carcinogenesis. Besides, hypoxic affected ATP production and transformatic capability of OTA could be investigated in further experiment [88].” (line 392-400, page 6 )
  19. “Nrf2 deficiency promote susceptible to OTA induced renal injury, thus modulation of Nrf2 provide therapeutic option to against OTA induced renal disease [89].” (line 404-406, page 6)
  20. “Cell death induced by OTA toxicity is concentration dependent manner. Besides, OTA is metabolized by P450(s) and OTA caused transcriptional suppression of glutathione S-transferase (GST) enzymes lead to nephrotoxicity by attenuate detoxification or conjugation of glutathione.” (line 408-411, page 6)
  21. “Meanwhile, organic anionic membrane transporter(s) are involved in the accumulation of OTA in the proximal tubule by evidence of intracellular OTA accumulation after organic anionic membrane transporter inhibitor (probenecid) was used. However, application of MPS is technical limitation to assess toxicology such as limited cell number, lack in throughput and limited in cell sorting [90].” (line 411-415, page 6)
  22. We rewrote the whole paragraph of prevention (line 433, page 9 to line 648, page 12)

  • As the authors pointed out in the review the mechanisms of OTA induced nephrotoxicity are complicated so it requires more efforts to comb, analyze and present scientific evidences. Otherwise, the review would not help readers to get a better understand about the topic.      

Ans: We classified the prevention and molecular mechanism induced by OTA. Besides, we added the future trends, directions or limitation of these studies, it would help readers to get a better understand the topic.

Round 2

Reviewer 2 Report

The authors have addressed reviewer's comments and revised the original manuscript accordingly. Therefore I would like to recommend the current version for publication. If needed, Dr. Hyun Jung Lee (hlee@uidaho.edu) can be invited to further assess the quality of the review.